# Sibship size, birth order and risk of asthma and allergy: protocol for a systematic review and meta-analysis

Daniil Lisik ,[1] Athina Ioannidou,[1] Gregorio Milani,[2,3] Sungkutu Nyassi,[1] Saliha Selin Özuygur Ermis,[4] Giulia C I Spolidoro,[3] Emma Goksör,[1] Göran Wennergren,[1,5] Bright I Nwaru[1]

¹Krefting Research Centre, University of Gothenburg, Gothenburg, Sweden
²Pediatric Unit, Fondazione IRCCS Ca' Granda Ospedale Maggiore Policlinico, Milan, Italy
³Department of Clinical Sciences and Community Health, University of Milan, Milan, Italy
⁴Department of Pulmonology, Dokuz Eylül University, Izmir, Turkey
⁵Department of Paediatrics, Sahlgrenska Academy at University of Gothenburg, Gothenburg, Sweden

**Correspondence to**
Daniil Lisik;
daniil.lisik@gmail.com

## ABSTRACT

**Introduction** The hygiene hypothesis suggests that reduced exposure to microbes might have contributed to the increase in prevalence and incidence of asthma and allergy observed during the second half of the last century. Following this proposal, several studies have investigated the role of sibship size and birth order in the development of asthma and allergic diseases, but the underlying evidence is conflicting. The objective of the present systematic review will be to identify, critically appraise and synthesise previous primary studies investigating the association of sibship size and birth order with the risk of asthma and allergic diseases.

**Methods and analysis** The following databases will be searched: AMED, CABI, CINAHL, Embase, Google Scholar, OAIster, Open Access Theses and Dissertations, Open Grey, ProQuest Dissertations & Theses Global, PsycINFO, PubMed, SciELO, Scopus, Web of Science and WHO Global Index Medicus. Studies published up until 31 December 2020 will be eligible. There will be no restrictions by language and geographical location. Risk of bias in the included studies will be assessed using the Effective Public Health Practice Project quality assessment tool. The produced evidence will be synthesised narratively, and studies that present comparable numerical data will be included in meta-analyses using random effects model.

**Ethics and dissemination** Only data from the published literature will be included in this systematic review. Therefore, no ethical approval is required. The final review paper will be published in a peer-reviewed journal.

**PROSPERO registration number** CRD42020207905.

## Strengths and limitations of this study

► This will be the first systematic review encompassing a comprehensive spectrum of the most common allergic and respiratory outcomes, in relation to sibship size and birth order.
► Inclusion of the leading databases, including search of the grey literature, enables a comprehensive identification of the relevant studies addressing the research question.
► The reproducibility of our work is enhanced through a priori outline of the review processes before the actual review starts.
► Self-reported diagnoses of the study outcomes are expected to make up a significant source of data from included studies, which gives the possibility of assessment bias.

## INTRODUCTION

The incidence and prevalence of asthma, along with allergic diseases such as allergic rhinitis and atopic eczema, were observed to have increased during the second half of the last century, in particular in the developed world.[1 2] More recent trends remain unclear, as both increase[3] and levelling off[1 4 5] have been suggested. Around 300 million people have asthma globally.[1] For allergic diseases, evidence indicates that there is still a global increase in prevalence.[5 6] Asthma and allergic diseases account for significant morbidity for individuals, as well as a substantial socioeconomic burden on the society.[2] Asthma results in roughly 14 million missed school days each year in the USA alone, and the morbidity is even higher for adults.[7] Allergic rhinitis is also associated with significant loss in productivity.[8] Furthermore, the WHO estimates that roughly 250 000 cases of death annually, worldwide, are due to asthma.[7] Identifying risk factors for asthma and allergy is therefore of great interest in order to reduce the burden associated with these diseases.

Over the last five decades, numerous hypotheses have been proposed to explain the observed increase in the prevalence of these diseases, a substantial part of the studies focusing on the role of environmental factors. One of the main hypotheses is the hygiene hypothesis, which was first proposed by Strachan[9] in 1989, and suggests that reduced microbial exposure during childhood increases the risk of developing asthma and allergy. One of the first proposed underlying biological mechanisms to the hygiene hypothesis was the observed

BMJ

skewing of balance towards T helper 2 cells (which have been associated with allergic sensitisation) in subjects lacking microbial stimuli and conversely a T helper 1 cell domination in subjects exposed to greater quantities of microbes.[10] Further research has broadened the explanatory model with additional factors, such as T regulatory cells and T helper 17 cells,[11] but the pathophysiology is yet to be fully understood.[12] Connected to the hygiene hypothesis is the proposed sibling effect, which suggests that the number of siblings and/or the birth order of a child in a family may play a role in the development of asthma and allergy, as a result of varying degrees of microbial exposure during childhood, depending on the number of siblings in total and/or the number of younger/older siblings.[13]

While several studies have investigated the association of sibship (group of individuals sharing the same pair of parents) size and birth order (the sequence in which members of a sibship are born) with risk of asthma and allergic diseases, findings are conflicting.[14] Karmaus and Botezan have estimated the proportion of cases attributable to the sibling effect to be 34% for atopic dermatitis, 56% for allergic rhinitis and 28% for asthma. Karmaus and Botezan have also argued that at least 30% of cases of asthma and allergy could be prevented if the causal factors for these conditions were better understood,[15] further indicating warranty for elucidating the sibling effect in relation to asthma and allergy. So far, there are no systematic reviews synthesising evidence from previous studies on the topic. A systematic synthesis of previous studies investigating the association of sibship size and birth order with risk of asthma and allergy will provide a clearer appreciation of the strength, magnitude and quality of the underlying evidence.

## Aim

To identify, critically appraise and synthesise previous primary studies investigating the association of sibship size and birth order with risk of asthma and allergic diseases.

## METHODS AND ANALYSES

This protocol is reported according to the recommendations of the Preferred Reporting Items for Systematic Review and Meta-Analysis Protocols (PRISMA-P),[16] which provides guidelines for a standardised, transparent and reproducible reporting of systematic review protocols. The PRISMA-P checklist is presented in online supplemental appendix 1. Updates to the protocol will be documented, and deviations from the protocol will be described in the final review paper. The protocol for this systematic review has been prospectively registered with the international prospective register of systematic reviews (PROSPERO, https://www.crd.york.ac.uk/prospero/display_record.php?RecordID=207905) with registration number CRD42020207905.

## Study eligibility criteria

### Study types and publication status

We will include observational epidemiological studies, including prospective and retrospective cohort studies, case–control studies and cross-sectional studies. Randomised controlled studies, quasirandomised controlled studies, controlled before-after studies and controlled clinical trials will not be considered, as interventional studies are not relevant for this research question. Animal studies, reviews, case studies, case series, expert opinions will also be excluded. Studies of any publication status will be eligible, and data used from them if available.

### Participants

Offspring of any age, gender, ethnic background and medical background, where the study context is that the participants are part of defined sibships. Studies with any amount of participants will be eligible.

### Exposures

Sibship size, birth order (number of older siblings) and number of younger siblings in the studied sibships.

### Outcome measures

Self-reported or objectively measured/diagnosed asthma and allergic disease in the sibships. For the purpose of encompassing all relevant literature on the topic, asthma and allergic disease will be defined broadly. Asthma will be defined as any type of asthma, including those based on symptom definition, such as wheezing, and those based on spirometry findings of variable expiratory airflow limitation.[17] Allergic disease will encompass any of the following: (A) allergic rhinitis/(rhino)conjunctivitis, food allergy, atopic eczema, urticaria, angio-oedema, anaphylaxis[18] and (B) indicators of hypersensitivity (and indirectly of allergic disease), which includes allergen-specific serum IgE test, skin prick test and provocation/challenge test. Conditions with primarily a genetic aetiology, such as hereditary angioedema,[19] will not be included in these definitions.

### Search methods

The search queries were developed using the PEO model: population, exposure and outcome (PEO). PEO is a specific implementation of population, intervention, control, outcome (PICO), used as a framework to produce effective search queries from formulated research questions, especially befitting retrieval of interventional and observational studies.[20] Since the population (P) will be defined broadly, that is, including both studies in children and adults, the actual search queries will be composed of two blocks: exposure (E) and outcome (O). A scoping search was performed in PubMed to identify previous studies on the topic and map relevant search terms. The search terms identified were: Medical Subject Headings (MeSH) and their corresponding alternatives in other databases, entry terms, free-text words and phrases. Subsequent scoping searches were made in PubMed with

Boolean operator 'NOT' between various MeSH and free-text terms, alternately, in order to identify more synonyms and related search terms. The developed search terms have been piloted and refined before they will be used to identify relevant studies. The search queries have been modified for each database to be searched in regards to, inter alia, support for controlled vocabulary and syntax. Peer Review of Electronic Search Strategies has been used to identify potential weaknesses in the search strategy. Details of the search strategy are presented in online supplemental appendix 2.

Studies will be retrieved from the following databases: AMED (via Ovid), CABI, CINAHL (via EBSCO), Embase (via Ovid), Google Scholar, PsycINFO (via ProQuest), PubMed, SciELO, Scopus, Web of Science and WHO Global Index Medicus. In addition, unpublished articles and grey literature will be retrieved through searches of OAIster, Open Access Theses and Dissertations, Open Grey and ProQuest Dissertations & Theses Global. Finally, studies will also be included from reference lists of the studies included in the review, as well as through contact with experts who have published in the field. All databases will be searched for articles published from inception of respective database up until 3 December 2020; an updated search will be performed at the completion of the review to ensure inclusion of studies published after the first search. There will be no language restrictions, and articles will be translated into English where possible. Articles that could not be translated will be reported in the final review paper. In Google Scholar, due to the fact that the amount of results is sometimes overwhelming, results will be retrieved from the first 300 hits.[21] Furthermore, the search query for Google Scholar has been significantly simplified, including only the most important terms in each block, due to an upper limit of 256 characters for search strings.[22]

## Data management
EndNote will be used for deduplication, full-text retrieval, secondary screening and for general management of retrieved studies. For primary screening, the articles will be imported to Rayyan QCRI.

## Screening/selection process
The first stage of screening will be based on the title and/or abstract of the articles. Articles that are clearly not relevant to the research question or clearly meet any of the exclusion criteria will be excluded. Articles, where there is doubt about relevancy, will be included to the next step. In the second stage of screening, the full text of the articles will be retrieved and assessed for eligibility. The reason for each article not being included will be documented and presented in a Preferred Reporting Items for Systematic Reviews and Meta-Analyses flow diagram in the final review paper.[23] The screening/selection will be independently performed by two reviewers. A third reviewer will arbitrate any disagreement.

## Data extraction
A data extraction form (online supplemental appendix 3) has been developed to extract data from included studies in a standardised and reproducible fashion. The form will be piloted and revised before being used in the review. If a study does not present needed data, authors of the study will be contacted. Extracted data will be presented in table form. The extraction will be independently conducted by two reviewers. A third reviewer will arbitrate any disagreement.

## Data items
The following data items will be summarised from each study: author of publication; country of origin of study; publication year; type of study design; sample size of study; source from where study participants were recruited; definition and assessment of sibship and birth order; duration of follow-up; confounding factors adjusted included in studies; study outcomes and their assessment; analysis methods; and main results.

## Quality assessment
Quality and risk of bias in the individual, included studies will be assessed using the Effective Public Health Practice Project Quality Assessment Tool (EPHPP).[24] The EPHPP contains six domains of assessment for each study, including study design, selection bias, confounding, blinding, study collection, withdrawals and dropouts. Based on the grading of each of the six domains, a global quality grading will be derived for each study. Detailed results will be presented in a separate table in the final review paper. Appraisal of quality and risk of bias will be independently performed by two reviewers. A third reviewer will arbitrate any disagreement.

## Data synthesis
Descriptive tables will be generated to present the key characteristics of the included studies. The produced evidence will be synthesised narratively. In addition, studies that present numerically comparable and reasonably homogenous (in terms of clinical and epidemiological settings of study participants) data will be synthesised quantitatively with meta-analyses in RevMan V.5 to produce pooled effect size estimates. Random effects model will be applied in the meta-analyses, because the included studies, solely based on published literature, are anticipated to not be similar in every aspect and thus do not estimate the same effect. This model is more conservative and provides a realistic scenario in the context of studies gathered solely from published literature.[25] Separate meta-analyses will be undertaken for each of the factors investigated (sibship size and birth order) in relation to each asthma and allergy outcome. The results of the meta-analyses will be presented in forest plots.

Risk ratio (RR) will be used as the outcome measure in the meta-analyses, because of its intuitive interpretative feature.[26] Data from studies presenting effect measures as OR, incidence rate ratio (IRR) or HR will be converted

to estimates of RR before combining with other studies, using the following formulas:

a. $RR \approx IRR$.

b. $RR \approx HR$ or $OR$ (if outcome is <15% by the end of follow-up).

c. $RR \approx \sqrt{OR}$ or $\frac{1 - 0.5^{\sqrt{HR}}}{1 - 0.5^{\sqrt{\frac{1}{HR}}}}$ (if outcome is ≥15% by the end of follow-up).[27]

Calculation of $I^2$ will quantify heterogeneity between the included studies.[28] Consideration will be taken, regarding that this statistic can be biased in meta-analyses with few studies.[29] Subgroup analysis will be performed to explore potential reasons for heterogeneity between studies with the following subgroup variables: (A) study design; (B) quality appraisal of studies; (C) classification of the study country into 'high-income', 'upper-middle-income', 'lower-middle-income' and 'low-income' economy, as defined by the World Bank[30]; (D) time during which the study was conducted, grouped into <1990, 1990–1999, 2000–2009 and 2010–2020; (E) participant age, grouped arbitrarily into <1 year, 1–6 years, 7–14 years and ≥15 years; and (F) gender, grouped into 'male' and 'female'. Subgroup analysis will be performed if there will be at least four (arbitrarily chosen cut-off[31]) studies in at least two subgroups. In addition, if more than 10 included studies present comparable numerical data,[32] meta-regression will be performed to explore the impact of explanatory variables (covariates) on the observed heterogeneity in estimates across studies.

To investigate whether the conclusions of the review are independent of arbitrary decisions, sensitivity analysis will be performed by only including studies that: (A) reach either 'strong' or 'moderate' global rating of quality in accordance to EPHPP and (B) have objectively verified diagnosis of asthma or allergic disease as outcome, with either ICD codes or verified medical examination as the basis for diagnosis. The sensitivity analysis will be reported in a summary table.

### Publication bias

Publication bias will be assessed with Funnel plot, as well as Begg's rank test and Egger's regression test,[33 34] with p<0.05 being defined as statistically significant. In case of (significant) publication bias, the trim-and-fill method will be implemented to analyse its influence on the review results.[35]

### Patient and public involvement

No patients or participants were involved in the development of this protocol or the design of this study.

### DISCUSSION

The conclusions that will potentially be drawn from this systematic review will be limited by the quality of the included studies. For this research question, the fact that all included studies will be observational limits the establishment of causality between sibship size, birth order and risk of asthma and allergic diseases.[36]

A strength of this study is the comprehensiveness of the search strategy, including 15 of the leading databases of formally published literature, as well as grey literature. There will be no restrictions in terms of language or geographical location. All these enable comprehensive identification of relevant studies for this research question. Furthermore, this systematic review will encompass a comprehensive spectrum of the most common asthma and allergic outcomes in relation to sibship size and birth order, thereby contributing to a broad overview of the existing evidence on the topic.

Asthma and allergic diseases pose a significant burden on both individuals and society. While the role of sibship size and birth order in the development of these diseases have been investigated in several studies, although with conflicting evidence, a systematic review of existing studies is essential in providing a clearer appreciation of the underlying evidence. This protocol presents the methodology to perform a comprehensive systematic review and meta-analysis of existing literature on the topic.

### ETHICS AND DISSEMINATION

Ethical approval will not be required due to the study being a systematic review of already published primary studies available in the public domain. Furthermore, patient consent will not be needed since data will stay aggregated.[37 38]

**Contributors** BIN conceived the study idea. DL wrote the protocol, with review and suggestions for improvement from BIN, EG, GW, and SSÖE. All authors commented and approved the last version being submitted.

**Funding** The authors have not declared a specific grant for this research from any funding agency in the public, commercial or not-for-profit sectors.

**Competing interests** None declared.

**Patient consent for publication** Not required.

**Provenance and peer review** Not commissioned; externally peer reviewed.

**ORCID iD**
Daniil Lisik http://orcid.org/0000-0002-0220-5961

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
