## [Reviewer comments · BMJ Open]

ARTICLE DETAILS

TITLE (PROVISIONAL)	Sibship Size, Birth Order and Risk of Asthma and Allergy: Protocol for a Systematic Review and Meta-Analysis
AUTHORS	Lisik, Daniil; Ioannidou, Athina; Milani, Gregorio; Nyassi, Sungkutu; Özüygür Ermis, Saliha Selin; Spolidoro, Giulia C. I.; Goksör, Emma; Wennergren, Göran; Nwaru, Brig

VERSION 1 – REVIEW

REVIEWER	Su, Hong Anhui Medical University
REVIEW RETURNED	14-Nov-2020

GENERAL COMMENTS	This manuscript is of interest. Overall, this study is well-designed. However, my main concerns are listed below: 1. It is well-known that environmental factors are closely related to the incidence of allergic disease, Could the author tell the readers the proportion of allergy cases attributable to the sibling effect? This will illustrate the importance of research.2. The author said” Connected to this hypothesis is the proposed sibling effect”. Sibling effect has been defined by: the number of older siblings, birth order, family size, the number of younger siblings and sibship size. Why the author only analyzes sibship size and birth order? Please illustrate.3.The author should give a relatively more specific introduction of the possible biological mechanisms between sibling effect and asthma and allergy.4. Inclusion and exclusion criteria should be clearly detailed. Did the included studies needed to include a certain minimum of participants? How did the authors manage duplicate record?5.Why not do the subgroup analysis by gender and age?6. Is $p < 0.05$ or others considered to be statistically significant in Begg's rank test and Egger's regression test? Please clearly state in the paper.
--

REVIEWER	Gundogdu, Zuhul Kocaeli Universitesi
REVIEW RETURNED	17-Nov-2020

GENERAL COMMENTS	Dear Author,
--------------

	This review is acceptable study for sibship and risk of allergy and asthma. But This is not new knowledge about this subject . However review subject could be about effect of sibship asthma on TTN. A number of studies have demonstrated that transient tachypnea of the newborn (TTN) is a risk factor for later asthma. Siblings' asthma was found to be independently associated with a diagnosis of TTN.(https://doi.org/10.1111/crj.13290)
--	--

REVIEWER	Cuomo, Barbara Operative Complex Unit of Pediatrics, Belcolle Hospital, pediatric department
REVIEW RETURNED	22-Feb-2021

GENERAL COMMENTS	You carefully plan a rigorous research methods with a detailed protocol to perform a good systematic review. The topic is relevant, considering clinical usefulness and the need for a comprehensive approach to allergy. Your search strategy protocol will be reported according to PRISMA and quality and risk of bias assessed using the EPHPP. The methodological plan provides guidance, structure, and transparency to the search methods, ensuring comprehensive and organized search methods. Participants and outcomes are described in detail, you clarify your clinical questions using PEO. Subgroups are carefully considered and prespecified in order to avoid data dredging. The criteria used for deciding which studies to include in the review are reported and matching the question to which you are seeking an answer. I really appreciate your choice to include grey literature, identifying all relevant evidence is an essential component and this can give important contributions to your ongoing systematic review. The Cochrane Handbook for Systematic Reviews of Interventions and the Institute of Medicine Standards for Systematic Review recommend incorporating grey literature in systematic reviews¹. Including all kind of literature in a researcher's search strategy may enrich the review's findings. I have just one suggestion, maybe you should specify that studies included in a meta-analysis might have not only comparable numerical data but also non-heterogeneous aspects. Finally I believe that a more specific diagnosis of asthma will help to exclude infection related symptoms. 1. Higgins J, Green S editors. Cochrane handbook for systematic reviews of interventions. 5.1.0 ed. Chichester, United Kingdom: The Cochrane Collaboration; 2011
---

VERSION 1 – AUTHOR RESPONSE

Reviewer 1

Dear Dr. Hong Su,

Thank you for your time and feedback.

Comment: It is well-known that environmental factors are closely related to the incidence of allergic disease, Could the author tell the readers the proportion of allergy cases attributable to the sibling effect? This will illustrate the importance of research.

Response: Study estimates concerning proportion of allergy and asthma cases attributable to the sibling affect have been appended (page 3-4 in Main Document)

Comment: The author said” Connected to this hypothesis is the proposed sibling effect”. Sibling effect has been defined by: the number of older siblings, birth order, family size, the number of younger siblings and sibship size. Why the author only analyzes sibship size and birth order? Please illustrate.

Response: We agree that the aspect of younger siblings should be investigated, as it is also considered an exposure component in the proposed sibling effect. We have added the exposure of younger siblings (page 3 and page 5 in Main Document). The rest of the listed exposures have already been included in "sibship size" (family size) and "birth order" (number of older siblings).

Comment: The author should give a relatively more specific introduction of the possible biological mechanisms between sibling effect and asthma and allergy.

Response: A more thorough theoretical background about the possible biological mechanisms has been appended (Page 3 in Main Document).

Comment: Inclusion and exclusion criteria should be clearly detailed. Did the included studies needed to include a certain minimum of participants? How did the authors manage duplicate record?

Response: We have decided on including studies with any amount of participants. This has been clarified in the main text (Page 5 in Main Document). Duplicate records will be removed in EndNote, as stated on Page 6 in Main Document.

Comment: Why not do the subgroup analysis by gender and age?

Response: We have added age and gender as subgroups, as this has been suggested to produce differing results worth investigating further (e.g. by Kikkawa et al, <https://doi.org/10.1111/cea.13100>). Although gender may pose ground for subgroup analysis of interest, we found few studies stratifying outcome data on gender during our scoping searches, making it less likely that there will be a satisfactory amount of studies to perform subgroup analysis on gender. However, as suggested, we have kept gender as one of the subgroups to investigate and, if we find insufficient studies, we will make an indication of this in the paper.

Comment: Is $p < 0.05$ or others considered to be statistically significant in Begg's rank test and Egger's regression test? Please clearly state in the paper.

Response: Defined p-value of statistical significane has been specified (Page 9 in Main Document).

Reviewer 2

Dear Dr. Zuhall Gundogdu,

Thank you for your time and feedback.

Comment: This review is acceptable study for sibship and risk of allergy and asthma. But This is not new knowledge about this subject . However review subject could be about effect of sibship asthma on TTN. A number of studies have demonstrated that transient tachypnea of the newborn (TTN) is a risk factor for later asthma. Siblings' asthma was found to be independently associated with a diagnosis of TTN.(<https://doi.org/10.1111/crj.13290>)

Response: To the best of our knowledge, this will be the first systematic review which encompasses a comprehensive spectrum of allergic conditions as well as asthma in relation to both sibship size and birth order. Previously, a limited number of reviews, such as by Karmaus and Botezan (<https://jech.bmj.com/content/56/3/209>), have either searched only one database, or investigated a substantially narrower selection of outcomes. Furthermore, there is a significant amount of recent primary studies which have not been synthesized, and there still is no clear consensus on the validity of the sibling effect in relation to asthma and allergy, warranting performing this comprehensive, systematic review of the existing evidences.

Reviewer 3

Dear Dr. Barbara Cuomo,

Thank you for your time and feedback.

Comment: I have just one suggestion, maybe you should specify that studies included in a meta-analysis might have not only comparable numerical data but also non-heterogeneous aspects.

Response: We have appended a statement regarding non-heterogeneity of data in meta-analyses (page 8 in Main Document).

Comment: Finally I believe that a more specific diagnosis of asthma will help to exclude infection related symptoms.

Response: We agree that a more specific diagnosis of asthma will help to exclude infection-related symptoms, e.g. the prevalent RSV bronchiolitis in early childhood. However, we stress that maintaining a wide definition-frame for this outcome will on the contrary assist us in our synthesis of the underlying evidence regarding asthma, as we aim to critically look at this aspect when analyzing the data stratified on age groups, quality appraisal of studies (as high-quality papers often attempt to minimize misdiagnosis) etc.